# Impact of Organic and Chemical Nitrogen Fertilizers on the Crop Yield and Fertilizer Use Efficiency of Soybean–Maize Intercropping Systems

Shifang Lin [1], Yijun Pi [1], Dayong Long [1], Jianjun Duan [2], Xingtao Zhu [3], Xiaoli Wang [1,*], Jin He [1] and Yonghe Zhu [4]

[1]  College of Agriculture, Guizhou University, Guiyang 550025, China
[2]  College of Tobacco Science, Guizhou University, Guiyang 550025, China
[3]  Guizhou Institute of Oil Crops, Guizhou Academy of Agriculture Sciences, Guiyang 550006, China
[4]  College of Agriculture, Nanjing Agricultural University, Nanjing 210095, China
[*]  Correspondence: xlwang@gzu.edu.cn

**Abstract:** The effect of the mixture (1:1) of chemical and organic nitrogen (N) fertilizer on crop yield quality and N fertilizer use efficiency remains elusive. A nitrogen field experiment was conducted in the growing seasons of 2020 and 2021 to investigate the effects of the mixture of chemical and organic N fertilizer on the crop yield, crop quality and nitrogen fertilizer use efficiency in a maize–soybean intercropping system in China. Four treatments applied at 150 kg N ha$^{-1}$ were used: no nitrogen fertilizer (CK), chemical N fertilizer (ChemF), mixture (1:1) of chemical and organic N fertilizer (ChemF + OrgF) and organic N fertilizer (OrgF). The results showed that the yield and aboveground N accumulation of both soybean and maize increased with the application of fertilizer. The ChemF + OrgF treatment had lower maize and soybean seed yields than for ChemF treatment, but higher than the other two treatments in both years, and the maize yield of the (ChemF + OrgF) treatment was significantly higher (14.9%) in 2021 than 2020. Yields were significantly positively correlated with aboveground N accumulation and fertilizer use efficiency, measured using the nitrogen partial productivity (NPP), nitrogen agronomic efficiency (NAE) and nitrogen fertilizer recovery rate (NFRR). The protein content tended to increase and the oil content tended to decrease under (ChemF + OrgF) applications in soybeans. The (ChemF + OrgF) treatment had the lowest starch content in maize. There was no significant difference in the nitrogen harvest index among treatments, while the NPP, NAE and NFRR were the highest for the application of chemical N fertilizer and significantly decreased with the addition of organic N fertilizer. We conclude that the mixture (1:1) of chemical and organic N fertilizer increased the seed yield and quality of maize, but only the seed yield of soybean.

**Keywords:** organic and chemical fertilizers; intercropping; yield and quality; N accumulation; fertilizer use efficiency

## 1. Introduction

Intercropping is a traditional planting mode and has been continuously addressed by agricultural scientists, with much research focusing on this topic worldwide [1–3]. Previous studies showed the advantages of intercropping to be mainly as follows: (i) inhibiting the growth of weeds, due to less sunlight penetrating compared to sole crops [4,5]; (ii) improving crop yield and quality (e.g., protein, oil and starch) [6–8]; (iii) improving crops' nutrient uptake and nitrogen use efficiency through facilitate interspecific [9–11]. Previous studies have shown that *Gramineae–Leguminosae* intercropping, especially soybean–maize, is the most widely used intercropping mode in the world [12]. More than 70% of intercropping systems used in China involve legume crops [13,14]. Maize being an exhaustive crop removes enough of the nutrients from the soil, and there is a need for taking soil replenishing

crops (legumes) in the cropping system. Hence, the maize legumes intercropping system is a more viable option for agricultural sustainability.

Nitrogen is an important nutrient element in farmland ecosystems and is an essential element for plant growth that is involved in plants' physiological and metabolic activities [15]. The inappropriate application of nitrogen fertilizer not only increases the environmental cost and pollution from agricultural production, but also reduces nitrogen use efficiency [16]. Optimizing nitrogen management, combined with a suitable planting system, can reduce chemical fertilizer input [17,18]. Reduction in the use of chemical fertilizer, combined with organic fertilizer, is a new measure to ensure good yields and soil health and requires more research [19]. Organic fertilizer contains a number of essential elements for crop growth, and can improve nitrogen use efficiency, and crop yield and quality [20,21]. A 20-year long-term field experiment showed that wheat yields for chemical fertilizer combined with organic fertilizer was the highest, followed by chemical fertilizer, and organic fertilizer was the lowest. Compared with chemical fertilizer, the combination of chemical and organic fertilizers increased the seed quality parameters (standard germination, seedling dry weight and seedling vigor) and the partial factor productivity of nitrogen, but decreased the nitrogen harvest index (NHI) and nitrogen use efficiency [22]. Combined chemical and organic fertilizer is more beneficial to plant growth and development than chemical fertilizer or organic fertilizer alone, with clearer effects on increasing yields and improving quality [23,24]. However, some studies showed no advantages of combined chemical and organic fertilizers in nitrogen uptake and partial productivity [25,26], or even reduced them [26]. These inconsistent results indicate that more work is needed to verify whether a reduction in chemical fertilizer combined with organic fertilizer could improve nitrogen fertilizer use efficiency and crop yield and quality.

Soybean–maize intercropping is a common planting pattern in southwest China [27]. A two-year field experiment was conducted to explore the effects of a mixture (1:1) of chemical and organic N fertilizer on crop yield, crop quality and nitrogen fertilizer use efficiency in a soybean–maize intercropping system.

## 2. Materials and Methods

### 2.1. Growth Conditions

The experimental site is located in Jichang Town (106°5'59" E, 26°6'29" N), Xixiu District, Anshun City, Guizhou Province, China. The elevation is 1271 m. This area has a humid subtropical monsoon climate, with average annual temperatures of 13.2–15.0 °C and rainfall of 968–1309 mm. The basic properties of the soil were pH 4.5, organic carbon 17.1 g kg$^{-1}$, available nitrogen 126.7 mg kg$^{-1}$, available phosphorus 20.9 mg kg$^{-1}$ and available potassium 159.5 mg kg$^{-1}$.

### 2.2. Materials and Design

The planting system is soybean–maize intercropping consisting of two rows of soybean and two rows of maize. The row spacing between soybean and maize was 60 cm, maize plant spacing was 30 cm (one seedling per hole) and soybean plant spacing was 20 cm (two seedlings per hole). In 2020, the maize hybrid variety was "Qiandan 16", and the soybean variety was "Anshun local spring soybean," The maize hybrid variety in 2021 was "Jinyu 908", and the soybean variety was "Fendou 97". The previous crop was ginger. Four fertilizer treatments—no nitrogen fertilizer (CK), chemical N fertilizer (ChemF), mixture (1:1) of chemical and organic N fertilizer (ChemF + OrgF), and organic N fertilizer (OrgF)—were applied at 150 kg N ha$^{-1}$, and the amount of phosphate fertilizer applied in each treatment was roughly the same—45 kg ha$^{-1}$. The organic N fertilizer (including N 2.3%) is produced by the Guizhou Wansheng Fertilizer factory. The nitrogen fertilizer is urea (including N 46.2%), and the phosphate fertilizer is calcium superphosphate (including $P_2O_5$ = 16%). All fertilizers were applied as basal fertilizer at one time before sowing. The alley between plots was 1 m, and the area of each plot was 22 m$^2$. The experiment used a

randomized block design, and each treatment was replicated three times, making a total of 12 plots.

### 2.3. Harvest and Measurement of Parameters

The maize and soybean were harvested when they reached the physiological maturity stage [28]. All maize and soybeans in each plot except the border plants were cut off by sickles (about 1 cm aboveground) and separated into straws and seeds. All seeds were dried at 80 °C for 48 h and weighed to determine the seed yield. A subsample (about 200 g of seeds) was randomly selected and stored before analysis. The straw was dried as above and weighed, then five representative soybean and five maize plants were chosen to determine the total nitrogen content. All samples were ground to fine power by a multi-function grinder (BJ-200, Baijie instrument Co., Ltd., Baijie, China).

The Kjeldahl method [29] was used to determine the total nitrogen content. About 0.5 g of fine power was digested with Concentrated sulfuric acid ($H_2SO_4$) and Hydrogen peroxide ($H_2O_2$). Then, the total nitrogen concentration was measured using a SKD-200 (Paiou Analytical instrument Co., Ltd., Shanghai, China). Protein was obtained by multiplying the total nitrogen content with the conversion coefficient (soybean: 5.71; maize: 6.25) [30]. Soluble sugar and starch were determined using anthrone colorimetry [31]. About 0.05 g of fine powder was extracted with $H_2SO_4$ and ethyl acetate reagent. Then, soluble sugar and starch were determined by spectrophotometry (UV755B, Youke Instrument Co, Ltd., Shanghai, China). The Soxhlet extraction method [30] was used to determine the oil content. About 2.0 g of fine power was extracted with ether on a crude fat tester (SZF-06G, Xinjia Electronics Co, Ltd., Xinjia, China).

Aboveground nitrogen accumulation (ANA, kg ha$^{-1}$) = seed nitrogen content (%) × seed dry weight (kg ha$^{-1}$)/100 + straw nitrogen content (%) × straw dry weight (kg ha$^{-1}$)/100.

NHI (%) = (seed nitrogen accumulation × 100)/above ground nitrogen accumulation.

Nitrogen partial productivity (NPP, kg k$^{-1}$) = crop yield/nitrogen application rate.

Nitrogen agronomic efficiency (NAE, kg kg$^{-1}$) = (yield difference between applying nitrogen and without applying nitrogen)/nitrogen application rate.

Nitrogen fertilizer recovery rate (NFRR, %) = (nitrogen uptake difference between with and without nitrogen application × 100)/nitrogen application rate.

### 2.4. Statistical Analysis

The differences between different treatments were analyzed using Excel 2010, ANOVA analysis and Origin2021 software (OriginLab Corporation, Northampton, MA, USA). The LSD method was used to test the significance of differences. Origin2021 software was used to draw correlation analysis figures.

## 3. Results

### 3.1. Yield and Seed Quality

The response of seed yields of soybean and maize were highest for the ChemF treatment and lowest for CK in 2020 and 2021 (Figure 1). The ChemF + OrgF treatment had the lowest seed yields of soybean and maize compared to the ChemF treatment, but it was higher than the other two treatments in both years. Additionally, the yield of maize treated with ChemF + OrgF in 2021 increased by 14.9% compared to that in 2020.

The soybean seed oil content in 2020 and protein content in 2021 showed no changes under the four treatments; the protein content in soybean seed under OrgF was higher than CK in 2020, and the oil content in soybean seed was lower than CK in 2021. There were no significant differences in the total protein and oil contents of soybean seed for the four treatments in 2020 and 2021 (Table 1).

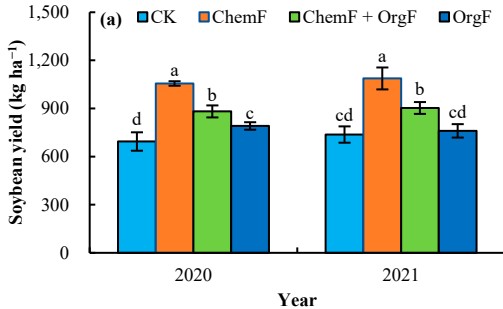 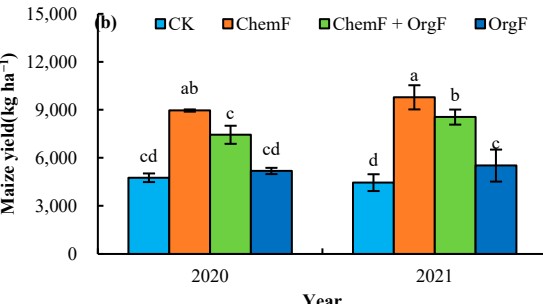

**Figure 1.** The seeds yield of soybean (**a**) and maize (**b**) with four treatments in soybean–maize intercropping system in 2020 and 2021. CK: no nitrogen fertilizer; ChemF: chemical N fertilizer; ChemF + OrgF: mixture (1:1) of chemical and organic N fertilizer; OrgF: organic N fertilizer. Different letters in each panel indicate significant differences at *p* = 0.05.

**Table 1.** The changes in protein content (PC), oil content (OC) and total protein–oil content (TP-OC) in soybean undergoing four treatments in 2020 and 2021. Different letters in the same column indicate significant differences at *p* = 0.05.

| Years | Treatments | PC/% | OC/% | TP-OC/% |
|---|---|---|---|---|
| 2020 | CK | 38 ± 1 b | 19 ± 0 a | 57 ± 1 a |
| | ChemF | 39 ± 0 ab | 19 ± 0 a | 57 ± 1 a |
| | ChemF + OrgF | 38 ± 0 b | 19 ± 1 a | 57 ± 1 a |
| | OrgF | 40 ±1 a | 18 ± 1 a | 59 ± 1 a |
| 2021 | CK | 43 ± 0 a | 20 ± 0 a | 63 ± 0 a |
| | ChemF | 42 ± 0 a | 20 ± 1 ab | 62 ± 1 a |
| | ChemF + OrgF | 43 ± 0 a | 19 ± 1 b | 61 ± 1 a |
| | OrgF | 43 ± 2 a | 18 ± 1 b | 62 ± 1 a |

Note: CK: no nitrogen fertilizer; ChemF: chemical N fertilizer; ChemF + OrgF: mixture (1:1) of chemical and organic N fertilizer; OrgF: organic N fertilizer. values are average ± SE. Different letters in the same column indicate significant differences at *p* = 0.05.

Compared with the ChemF treatment, ChemF + OrgF significantly increased the maize starch content (by 9.2% in 2010 and 8.4% in 2021). There were no significant differences in maize protein and soluble sugar contents between 2020 and 2021 (Table 2).

**Table 2.** The changes in protein content (PC), starch content (SC) and soluble sugar content (SSC) in maize with four treatments in 2020 and 2021. Different letters in the same column indicate significant differences at *p* = 0.05.

| Years | Treatments | PC (%) | SC (%) | SSC (%) |
|---|---|---|---|---|
| 2020 | CK | 10 ± 0 b | 23 ± 0 c | 7 ± 0 a |
| | ChemF | 10 ± 0 b | 25 ± 1 b | 8 ± 1 a |
| | ChemF + OrgF | 12 ± 1 a | 28 ± 2 a | 7 ± 1 a |
| | OrgF | 10 ± 0 ab | 26 ± 1 b | 7 ± 0 a |
| 2021 | CK | 13 ± 1 a | 24 ± 1 c | 9 ± 1 a |
| | ChemF | 14 ± 1 a | 25 ± 1 bc | 10 ± 1 a |
| | ChemF + OrgF | 14 ± 0 a | 27 ± 1 a | 9 ± 2 a |
| | OrgF | 13 ± 0 a | 26 ± 1 ab | 8 ± 1 a |

Note: CK: no nitrogen fertilizer; ChemF: chemical N fertilizer; ChemF + OrgF: mixture (1:1) of chemical and organic N fertilizer; OrgF: organic N fertilizer. values are average ± SE. Different letters in the same column indicate significant differences at *p* = 0.05.

### 3.2. Aboveground Nitrogen Accumulation

The ANA of maize and soybean was the highest for the ChemF treatment and the lowest for CK in 2020 and 2021. The ANA of soybean under ChemF + OrgF was significantly

lower than that of ChemF treatment, while the that of maize under ChemF + OrgF was the same as that for ChemF (Figure 2).

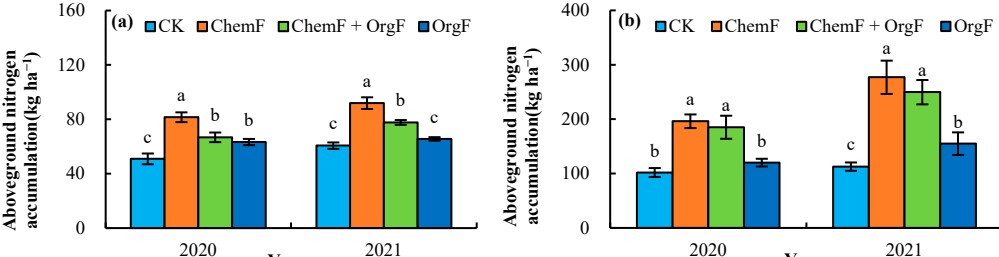

**Figure 2.** Aboveground nitrogen accumulation under four treatments in soybean (**a**) and maize (**b**) in 2020 and 2021. CK: no nitrogen fertilizer; ChemF: chemical N fertilizer; ChemF + OrgF: mixture (1:1) of chemical and organic N fertilizer; OrgF: organic N fertilizer. Different letters above columns within each panel indicate significant differences at $p = 0.05$.

### 3.3. Nitrogen Fertilizer Use Efficiency

The NHI showed no difference among the four treatments in soybean and maize in both years (Table 3). The NPP, NAE and NFRR were the lowest in the OrgF treatment, but the highest for the ChemF treatment in both soybean and maize, although soybean had a significantly lower NPP, NAE and NFRR than maize (Table 3).

**Table 3.** Nitrogen harvest index (NHI), nitrogen partial productivity (NPP), nitrogen agronomic efficiency (NAE) and nitrogen fertilizer recovery rate (NFRR) in soybean in 2020 and 2021.

|  | Years | Treatments | NHI (%) | NPP (kg kg$^{-1}$) | NAE (kg kg$^{-1}$) | NFRR (%) |
|---|---|---|---|---|---|---|
| Soybean | 2020 | CK | 89 ± 2 a | — | — | — |
|  |  | ChemF | 89 ± 1 a | 7 ± 0 a | 2 ± 0 a | 20 ± 2 a |
|  |  | ChemF + OrgF | 88 ± 0 a | 6 ± 0 b | 1 ± 0 b | 11 ± 2 b |
|  |  | OrgF | 88 ± 2 a | 5 ± 0 c | 1 ± 0 c | 8 ± 2 b |
|  | 2021 | CK | 91 ± 1 a | — | — | — |
|  |  | ChemF | 88 ± 1 a | 7 ± 1 a | 2 ± 1 a | 21 ± 3 a |
|  |  | ChemF + OrgF | 87 ± 1 a | 6 ± 1 b | 1 ± 0 b | 11 ± 1 b |
|  |  | OrgF | 88 ± 3 a | 5 ± 0 c | 0 ± 0 c | 3 ± 1 c |
| Maize | 2020 | CK | 72 ± 0 a | — | — | — |
|  |  | ChemF | 71 ± 3 a | 60 ± 4 a | 28 ± 4 a | 63 ± 8 a |
|  |  | ChemF + OrgF | 75 ± 6 a | 50 ± 4 b | 18 ± 4 b | 56 ± 14 a |
|  |  | OrgF | 72 ± 5 a | 35 ± 1 c | 3 ± 1 c | 12 ± 5 b |
|  | 2021 | CK | 79 ± 2 a | — | — | — |
|  |  | ChemF | 77 ± 5 a | 65 ± 5 a | 36 ± 5 a | 119 ± 20 a |
|  |  | ChemF + OrgF | 78 ± 3 a | 57 ± 3 a | 27 ± 3 b | 91 ± 15 a |
|  |  | OrgF | 74 ± 5 a | 37 ± 7 b | 10 ± 3 c | 28 ± 14 b |

Note: CK: no nitrogen fertilizer; ChemF: chemical N fertilizer; ChemF + OrgF: mixture (1:1) of chemical and organic N fertilizer; OrgF: organic N fertilizer. values are average ± SE. Different letters in the same column indicate significant differences at $p = 0.05$.

### 3.4. Correlation Analysis

The seed yields of maize and soybean were significantly positively correlated with ANA and nitrogen fertilizer use, but not with NHI and quality (Figure 3). The ANA was positively correlated with NPP, NAE and NFRR, indicating that increases in nitrogen accumulation could improve nitrogen fertilizer use efficiency.

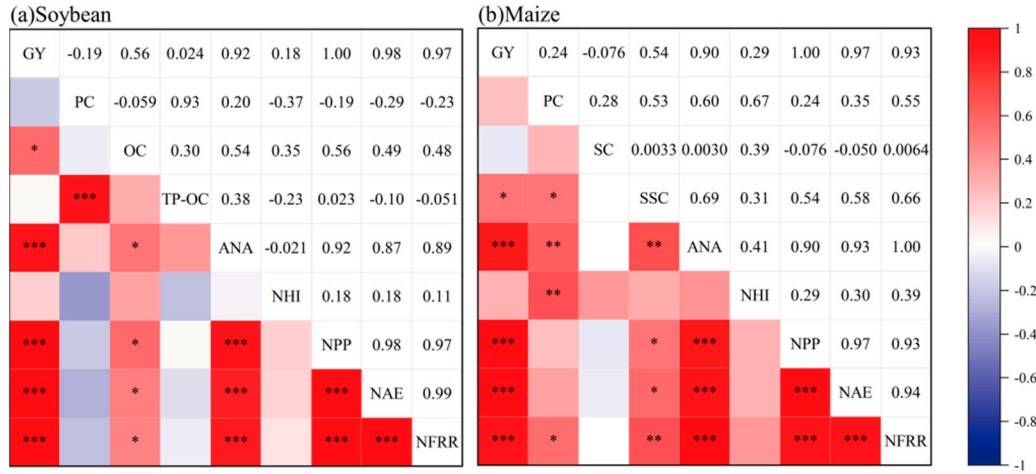

**Figure 3.** Correlation matrix between seed yield (SY), seed quality [soybean: protein content (PC), oil content (OC), total protein–oil content (TP-OC); maize: protein content (PC), starch content (SC), soluble sugar content (SSC)], aboveground nitrogen accumulation (ANA), nitrogen harvest index (NHI), nitrogen partial productivity (NPP), nitrogen agronomic efficiency (NAE) and nitrogen fertilizer recovery rate (NFRR) in soybean (**a**) and maize (**b**). The figures show that the correlation coefficient. * $p < 0.05$, ** $p < 0.01$, *** $p < 0.001$.

## 4. Discussion

### 4.1. Response of the Seed Yield and Quality to Different Fertilizer Treatments

There are many factors affecting crop yields, including planting density, cultivation measures, tillage methods, field management and fertilization treatment [32]. The use of fertilizers such as chemical and organic fertilizers could significantly increase crop yields by changing soil nutrient levels [33,34]. The results of this two-year field experiment showed that chemical N fertilization increased the yields of soybean and maize in a maize–soybean intercropping system, but with no effect from organic N fertilizer. The yield of soybean and maize with organic N fertilizer was significantly lower than for a mixture (1:1) of chemical and organic N fertilizer and chemical N fertilizer. The reason was that the rapid release of nitrogen and phosphorus in chemical fertilizers directly replenished the soil nutrients, while the nutrients in bio-organic fertilizers can only be absorbed by crops after transformation, which is a long process. Thus, the effect of organic fertilizer on crop seed yields was not significant in the short term. In 2021, compared with 2020, the soybean yield of each treatment was relatively stable, and maize yields were significantly increased under ChemF + OrgF treatment. This result is not completely consistent with the results of Li et al. and Zhang et al. [35,36], which can be explained by the soil priming effect of nitrogen additions [37]. Moreover, we found that both the soybean and maize yield improvements were associated with ANA, indicating the essential role of nitrogen accumulation in yield performance in different crops [28,29,38,39]. The higher accumulation of aboveground nitrogen may be caused by the high root growth [40] and/or the high nitrogen availability with chemical fertilizer applications [39], which should be verified in the near future.

Crop quality is controlled by the crop genes, as well as external environmental factors and cultivation conditions [41]. Few studies have focused on the effects of chemical fertilizer reduction and organic fertilizer on the quality of soybean and maize, and the results from different studies are inconsistent. A study showed that a mixture of chemical and organic N fertilizer improved the quality of soybean and maize [42], although other studies [43,44] showed that with fertilizer, the increase in one quality index will also cause an imbalance in other quality indexes. In our studies, we found that the application of chemical N fertilizer had no significant effect on soybean quality, but with the application of organic N fertilizer, the protein and the oil contents decreased, and the total protein–oil content did not significantly change. Fertilization had no significant effect on the protein and soluble sugar contents of maize, but the mixture (1:1) of chemical and organic N fertilizer significantly

increased the starch content. This is similar to the results of previous studies [44,45]. The responses of various quality indexes of soybean and maize to the reduction in chemical fertilizer and the application of organic fertilizer were not consistent. The possible reason was that total protein–oil content of soybean is mainly determined by genotype, and the decomposition of organic fertilizer can release organic acid that can promote the absorption of nutrients by crops, promote the metabolic process of microorganisms and increase the contents of protein and starch in crop seeds [46], but organic fertilizers release a lot of heat in the process of decomposition. In addition, the factors affecting crop yield and quality are numerous and complicated, and soil microorganisms and the soil environment may also be important [47,48].

### 4.2. Effects of Different Fertilizer Treatments on Nitrogen Fertilizer Use Efficiency

The commonly used indicators for nitrogen fertilizer use efficiency are NHI, NPP, NAE and NFRR. The results showed that the nitrogen fertilizer use efficiency of soybean and maize was consistent, with no significant difference in NHI among treatments, and the highest NPP, NAE and NFRR were under the ChemF treatment, which tended to decrease significantly with the reduction in chemical N fertilizer, with highly significant positive correlations between NPP, NAE and NFRR. This is inconsistent with the results of Dong et al. and Wang et al. [40,49]. The possible reason is because the soil organic nitrogen must be mineralized into inorganic nitrogen before it can be absorbed and used by crops [50], while the organic fertilizer we supplied has a high carbon/nitrogen ratio and can reduce organic nitrogen mineralization [51]. Therefore, even with the same amount of nitrogen application, the nitrogen fertilizer use efficiency was highest in the ChemF and lowest in the OrgF treatment. The reason for the lack of significant differences in NHI among the treatments may be that NHI was not related to fertilizer application but only to nitrogen accumulation in the crop itself, which should be verified in the near future. We also found that the yield was significantly positively correlated with NPP, NAE and NFRR, which indicated that high yields would promote high nitrogen fertilizer use efficiency. Since this study only measured the indexes for two consecutive years, which is a short period of time, subsequent studies would consider the long-term nutrient balance and follow up the dynamic changes of crop yield quality and nitrogen fertilizer use efficiency between years in the long term under rationing conditions, with a view to formulating a fertilizer management strategy for sustainable production.

### 5. Conclusions

The seed yield, ANA and nitrogen use efficiency of both soybean and maize for the mixture (1:1) of chemical and organic N fertilizer was lower than for the application of chemical N fertilizer, but the mixture (1:1) of chemical and organic N fertilizer significantly increased starch content of maize. There was a positive correlation between seed yield and ANA and nitrogen use fertilizer efficiency. High nitrogen accumulation could contribute to the high nitrogen use efficiency. The effects of the different fertilizer schemes on soil parameters need to be verified in the future.

**Author Contributions:** Conceptualization, X.W. and J.H.; formal analysis, S.L., Y.P., X.W. and J.H.; funding acquisition, X.W.; investigation, S.L., Y.P., D.L., J.D. and X.Z.; methodology, S.L., X.W. and J.H.; supervision, J.D., X.W. and J.H.; writing—original draft, S.L., X.W. and J.H.; writing—review and editing, Y.P., D.L., J.D., X.Z., X.W., J.H. and Y.Z.; All authors have read and agreed to the published version of the manuscript.

**Funding:** This research was supported by the National Natural Science Foundation of China (31860339).

**Institutional Review Board Statement:** Not applicable.

**Data Availability Statement:** Not applicable.

**Conflicts of Interest:** The authors declare no conflict of interest.

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
