# Peer review of "Impact of Organic and Chemical Nitrogen Fertilizers on the Crop Yield and Fertilizer Use Efficiency of Soybean–Maize Intercropping Systems"

_agriculture, doi:10.3390/agriculture12091428_

Round 1
Reviewer 1 Report
There were some major revisions regarding the material and methods which is mentioned in the main file

Author Response
Response to Reviewer 1 Comments
- Title:
Please change the title: Impact of Organic and Chimical Fertilizers on crop Yield and Fertilizer Use Efficiency of Soybean - Maize Intercropping System
Response: Accepted. We have revised the title.
Abstract:
The first sentence or two sentences should be introductory highlighting the need of the study. Please rewrite.
Response: Accepted. We have rewrite the sentence.
Keywords:
Rewrite. Use more appropriate words, not the lengthy phrase.
Response: Accepted. We have rewrite this part.
Introduction:
Also mention the following in the introduction part:
Maize being an exhaustive crop removes enough of nutrients from the soil and there is a need for taking soil replenishing crops (legumes) in the cropping system. Hence, maize legumes intercropping system system is a more viable option for agricultural sustainability.
As you have mentioned that combined application of organic and chemical nutrients showed contradictory results earlier. Now, please justify, how a two-year field study can conclude it.
Response: We only reported the data collected in two years but the experiment will continue to conducted.
Materials and methods:
Please provide monthly/ fortnightly meteorological data (temperature, RH, rainfall, sunshine hour/day) of the growing location for the growing period graphically.
Response: The data we have was added in the manuscript. .
Whether the maize is hybrid or variety? Mention clearly and everywhere.
Please mention which cropping system was adopted earlier before commencement of the study?
Response: Accepted.We have revised accordingly and highlight in yellow in materials and methods.
“All fertilizers were applied as basal fertilizer at one time before sowing.” That means, nod topdressing of N. If so, why?
Response: The experiment is a long-term positioning experiment, and all fertilizers are applied at once.
Conclusions:
Rewrite it by highlighting the future scope of research.
Response: The information you required was added.
References
Follow the MDPI style.
Response: Accepted. We have revised accordingly.
Reviewer 2 Report
This article aimed to explore combining chemical and organic N fertilizers in intercropping. The experimental description is simple. Yields and some N parameters were evaluated. With the amount of data reported, this should be a short communication.
As the manuscript was not numbered, I have appended my comments on the attached PDF (Don't miss it). My major points include:
1. The source of N fertilizer should be clear.
2. The abbreviations are odd. N is strictly for nitrogen and should not be used to represent anything else. Too many confusing abbreviations made it difficult to follow.
3. The comparison of how the mixed fertilizer relative to others should be stressed
4. The word "reduction" in the mixed fertilizer is confusing because the rates were all the same. This should be changed in the title too.
5. The roles of soybean in intercrop with maize were not discussed.
6. The authors should combine their R&D as they kept repeating the results in the discussion section.
7. If the soil data can be added that will be a plus to this study to know the residual N for the treatments.
8. Statistical section should be improved to reflect things shown in the result section.
9. Conclusion is very weak
Goodluck

Author Response
Response to Reviewer 2 Comments
- The source of N fertilizer should be clear.
Response: Accepted.We have added the source of N fertilizer.
- The abbreviations are odd. N is strictly for nitrogen and should not be used to represent anything else. Too many confusing abbreviations made it difficult to follow.
Response: Accepted. We have revised accordingly.
- The comparison of how the mixed fertilizer relative to others should be stressed
Response: Only one mixed fertilizer was applied in the experiment.
- The word "reduction" in the mixed fertilizer is confusing because the rates were all the same. This should be changed in the title too.
Response: Accepted. We have revised accordingly.
- The roles of soybean in intercrop with maize were not discussed.
Response: In the experiment, the crop planting mode was soybean-corn intercropping, and soybean played the same role in maize intercropping in each treatment.
- The authors should combine their R&D as they kept repeating the results in the discussion section.
Response: Accepted.We have revised accordingly and highlight in yellow in discussion.
- If the soil data can be added that will be a plus to this study to know the residual N for the treatments.
Response: We do not measure the soil data but we will do it in the future.
- Statistical section should be improved to reflect things shown in the result section.
Response: Accepted.We have revised accordingly and highlight in yellow in statistical section.
- Conclusion is very weak
Response: Accepted. We have revised accordingly.
Round 2
Reviewer 1 Report
I would like to thank the authors for the relevant improvements. In my opinion this manuscript can be published in its present form
Author Response
I would like to thank the authors for the relevant improvements. In my opinion this manuscript can be published in its present form
Response: Thank you for your positive comments and thanks for your useful comments again.
Reviewer 2 Report
Dear authors,
This is a great improvement to the first draft. I have included my comments in the attached (DO NOT MISS THE ATTACHMENT) revised manuscript.
1. Consider removing "1/2" from the mixed fertilizer abbreviations as it does not add or change the abbreviation meaning. This should be considered in the tables and figures too.
2. Consider using ratio (1:1) instead of using equal rate which can imply something else for the mixed fertilizer
3. Round the average in Table 4 to 0 decimal place
4. Consider removing dot "." in all your kg.kg-1
5. Copy the footnote "values are average ±SE" to all your tables where appropriate
I hope these suggestions help.

Author Response
- Consider removing "1/2" from the mixed fertilizer abbreviations as it does not add or change the abbreviation meaning. This should be considered in the tables and figures too.
Response: Accepted. Thanks for your useful suggestion, we have corrected in whole manuscript.
- Consider using ratio (1:1) instead of using equal rate which can imply something else for the mixed fertilizer
Response: Accepted. We have corrected in whole manuscript.
- Round the average in Table 4 to 0 decimal place
Response: Accepted. We have revised accordingly and highlight in yellow.
- Consider removing dot "." in all your kg.kg-1
Response: Accepted. We have deleted the dot.
- Copy the footnote "values are average ±SE" to all your tables where appropriate
Response: Accepted. We have corrected and applied it for all tables.